# Development of the Rabbit NASH Model Resembling Human NASH and Atherosclerosis

**DOI:** 10.3390/biomedicines11020384

**Published:** 2023-01-27

**Authors:** Momoko Hayashi, Yoshibumi Kuwabara, Kuniji Ito, Yoshiaki Hojo, Fumiaki Arai, Kazuki Kamijima, Masakazu Takeiri, Xiaojing Wang, Pan Diao, Jun Nakayama, Naoki Tanaka

**Affiliations:** 1Department of Metabolic Regulation, Shinshu University School of Medicine, Matsumoto 390-8621, Japan; 2Kitayama Labes Co., Ltd., Ina 396-0025, Japan; 3Department of Gastroenterology, Lishui Hospital, Zhejiang University School of Medicine, Lishui 310030, China; 4Department of Molecular Pathology, Shinshu University School of Medicine, Matsumoto 390-8621, Japan; 5Department of Global Medical Research Promotion, Shinshu University Graduate School of Medicine, Matsumoto 390-8621, Japan; 6International Relations Office, Shinshu University School of Medicine, Matsumoto 390-8621, Japan; 7Research Center for Social Systems, Shinshu University, Matsumoto 390-8621, Japan

**Keywords:** animal model, NASH, liver fibrosis, atherosclerosis, cholesterol, ER stress

## Abstract

Non-alcoholic steatohepatitis (NASH) is a chronic liver disease which may progress into liver fibrosis and cancer. Since NASH patients have a high prevalence of atherosclerosis and ensuing cardiovascular diseases, simultaneous management of NASH and atherosclerosis is required. Currently, rodents are the most common animal models for NASH and accompanying liver fibrosis, but there are great differences in lipoprotein profiles between rodents and humans, which makes it difficult to reproduce the pathology of NASH patients with atherosclerosis. Rabbits can be a promising candidate for assessing NASH and atherosclerosis because lipoprotein metabolism is more similar to humans compared with rodents. To develop the NASH model using rabbits, we treated the Japanese White rabbit with a newly developed high-fat high-cholesterol diet (HFHCD) containing palm oil 7.5%, cholesterol 0.5%, and ferrous citrate 0.5% for 16 weeks. HFHCD-fed rabbits exhibited NASH at 8 weeks after commencing the treatment and developed advanced fibrosis by the 14th week of treatment. In addition to hypercholesterolemia, atherosclerotic lesion developed in the aorta after 8 weeks. Therefore, this rabbit NASH model might contribute to exploring the concurrent treatment options for human NASH and atherosclerosis.

## 1. Introduction

Non-alcoholic fatty liver disease (NAFLD) is a common liver disease in Western countries, ranging from simple steatosis to steatohepatitis [1,2,3,4]. Non-alcoholic steatohepatitis (NASH) is characterized by hepatic inflammation and hepatocyte degeneration in addition to macrovesicular steatosis, and may progress into liver fibrosis, cirrhosis, and hepatocellular carcinoma [1,2,3,4]. The pathogenesis of NAFLD includes systemic metabolic abnormalities, such as dyslipidemia, excessive fat accumulation in adipose tissue, and insulin resistance, and can be deemed a hepatic phenotype of metabolic syndrome [1,2,3,4]. The molecular mechanisms for the development and progression of NAFLD/NASH are highly complicated and have been recognized as “multiple-hit theory”. For example, overload of free cholesterol (F-Chol) or palmitate increases endoplasmic reticulum (ER) stress and oxidative stress in the hepatocytes, leading to inflammasome activation, lipoapoptosis, and pyroptosis [5,6,7,8,9,10]. 

The liver is a central organ for the systemic lipid metabolism, and lipoproteins play a key role in the transport of lipids between the liver and peripheral tissues. Very low-density lipoprotein (VLDL) is involved in the efflux of fatty acids from the liver to the peripheral tissues, and high-density lipoprotein (HDL) has the function of eliminating excessive cholesterol from peripheral tissues, designated as reverse cholesterol transport (RCT) system. The imbalance in lipoprotein metabolism contributes to the pathology of NAFLD/NASH and atherosclerosis [11,12,13]. 

One of the most common causes of death in NAFLD/NASH patients is cardiovascular disease (CVD) [14]. Atherosclerosis is the risk factor for CVD, and NAFLD patients have a high score on intima–media thickness of the carotid artery and a higher prevalence of hypertension [14,15,16]. Numerous reports indicate that NAFLD is an independent risk factor for CVD, and the relationship between NAFLD and cerebrovascular disease has also been pointed out [17,18,19]. Thus, concurrent treatment for NAFLD and atherosclerosis is an important issue. 

Since the effective therapy for NASH and ensuing fibrosis has not been established yet, its development is an urgent problem. Rodents, such as mice and rats, are widely used as animal models for NASH, due to easy handling and management. To elucidate the development of NASH and develop therapeutic agents, numerous animal models were generated, such as genetic models (e.g., *ob/ob* mice, *db/db* mice, and *Apoe*-null mice), dietary models (e.g., methionine- and choline-deficient diet, and high-fat high-sucrose diet), as well as their combinations [20,21,22,23,24,25]. However, there are some species differences between these rodents and humans in several aspects, including lipoprotein composition and bile acid (BA) metabolism. For instance, rats have no gallbladder, and the dominant composition of BA in mice is hydrophilic but hydrophobic in humans [26,27]. Furthermore, mice have high levels of HDL in their lipoprotein composition and exhibit a high low-density lipoprotein (LDL) receptor (LDLR) activity. The VLDL component in mice and rats includes apolipoprotein B (apoB)-48 and apoB-100, but only apoB-100 in humans [28]. These differences may lead to the poor reproducibility of preclinical results using the NASH animal models for human clinical trials, becoming an obstacle to therapeutic development. Since rabbits have compositions that are similar to human lipoproteins and BAs [29,30], they are expected to mimic the human NASH pathogenesis and pharmacological response to the therapeutic agents. 

Additionally, mice and rabbits are commonly used for investigating the pathogenesis of atherosclerosis. Human atherosclerotic plaque is composed of cholesterol-laden macrophages (foam cells), proliferated vascular smooth muscle cells and fibroblasts, and lipids released from dead foam cells (necrotic core) [31]. Administration of a high-cholesterol diet to mice and rabbits disrupting LDLR leads to the development of atherosclerotic plaques [32,33]. The mice exhibit some pathological features similar to human atherosclerotic lesions, such as necrotic core and lipid-laden foam cells. However, in contrast to humans, the plaques are stable, rarely rupture spontaneously and do not represent CVD episodes, such as myocardial infarction (MI) and stroke [32]. Watanabe heritable hyperlipidemic rabbits, which develop hypercholesterolemia and advanced atherosclerosis due to LDLR deficiency, exhibit more similarities to human atherosclerotic plaques, such as calcification and thin fibrous cap, which are susceptible to plaque rupture and ensuing MI [34,35]. These differences between mice and rabbits prompted us to speculate regarding the usefulness of rabbit models for evaluating the human pathology of atherosclerosis. 

To obtain good extrapolation to human NASH/atherosclerosis, animal models more closely resembling human pathologies and metabolic systems are desired. Therefore, we tried to develop the rabbit NASH/atherosclerosis model using a newly developed high-fat high-cholesterol diet (HFHCD) and Japanese White rabbits. 

## 2. Materials and Methods

### 2.1. Animals and Treatment

Pathogen-free male 11-week-old Japanese White rabbits Kbs:JW (Kitayama Labes Co., Ltd., Nagano, Japan) were housed at a temperature of 17–28 °C and humidity of 40–90% in a 12-hour light/dark cycle. The rabbits were fed a standard diet (n = 3, LRC4 obtained from Oriental Yeast Co., Ltd., Tokyo, Japan) or a HFHCD (n = 3, LRC4 91.5%, palm oil 7.5%, cholesterol 0.5%, and ferrous citrate 0.5%) with free access to sterilized water for up to 16 weeks. The composition of LRC4 is listed in Appendix A. A time–course study was also conducted under the same conditions in the control group (12 weeks, n = 3), and in the 4-week (n = 3), 8-week (n = 4), and 14-week (n = 4) groups. Routine blood sampling was performed during the trial period. After each test period, rabbits were anesthetized and sacrificed for blood and tissue sampling. The blood samples were collected from the cardiac cavity and maintained at −80 °C until the assay. Each organ was harvested, weighed, and the parts of the organs were rapidly frozen with liquid nitrogen for subsequent assays. The liver and aorta were fixed with 10% formalin neutral buffer solution (FUJIFILM Wako Pure Chemical Corporation, Osaka, Japan) for histological analysis. All animal experiments were conducted in accordance with the National Academy of Sciences. The animal study protocol [#020042 (15 October 2020)] was approved by the Shinshu University School of Medicine “Guide to the Care and Use of Experimental Animals”.

### 2.2. Histological Analysis

Formalin-fixed liver tissues and aorta were embedded in paraffin, cut into 3-μm thick sections, and stained with hematoxylin and eosin (HE) or Azan–Mallory method. 

### 2.3. Biochemical Analysis

Serum alanine aminotransferase (ALT), aspartate aminotransferase (AST), and serum/liver triacylglycerol (TG), non-esterified fatty acid (NEFA), total cholesterol (T-Chol), F-Chol, phospholipid (PL), and total bile acid (TBA) were detected using commercial enzyme assay kits (FUJIFILM Wako Pure Chemical Corporation, Osaka, Japan). Total liver lipid extraction is conducted using the hexane:isopropanol method as previously described [36] and quantified using the same kits. 

### 2.4. Quantification of mRNA Measurement

Liver total RNA was extracted with NucleoSpin RNA plus (Macherey-Nagel GmbH & Co. KG, Düren, Germany) and reverse-transcribed using PrimeScript RT Reagent kit (Takara Bio Inc., Shiga, Japan). The mRNA levels were detected via real-time PCR using THUNDERBIRD SYBR qPCR Mix (Toyobo Co., Ltd., Osaka, Japan) on QuantStudio 3 Real-Time PCR System (Thermo Fisher Scientific, Waltham, MA) [37,38,39,40]. The primer pairs are listed in Appendix A. Glyceraldehyde 3-phosphate dehydrogenase (GAPDH) was used as an internal control gene.

### 2.5. Western Blot Analysis

Liver tissues were homogenized with lysis buffer (Sucrose 0.25 M, Tris chloride 25 mM, KCl 25 mM, MgCl_2_ 5 mM, 0.5% (*w*/*v*) Triton X, pH 7.4) containing protease and phosphatase inhibitor cocktail (Thermo Fisher Scientific, Waltham, MA, USA). Protein concentrations were measured colorimetrically using a Pierce BCA Protein Assay kit (Thermo Fisher Scientific, Waltham, MA, USA). Liver total protein samples were loaded in each lane (30 µg/lane) and separated by 10% sodium dodecyl sulfate–polyacrylamide gel electrophoresis (SDS-PAGE). Following SDS-PAGE, the proteins were transferred to polyvinylidene difluoride membranes (Immobilin-P, Merck KGaA, Darmstadt, Germany) and the membranes were blocked with 3% non-fat dry milk for 1 hour at room temperature and incubated with primary antibodies overnight for 4 °C. After washing, the membranes were exposed to horseradish peroxidase-conjugated secondary antibody (115-035-003, Jackson ImmunoResearch Laboratories, West Grove, PA) for 1 hour at room temperature. Specific bands were detected using ECL Select Western Blotting Detection Reagent (Cytiva, Tokyo, Japan) on ChemiDoc Touch Imaging System (Bio-Rad Laboratories, Inc., Hercules, CA, USA). Band intensities were quantified using the manufacture’s software (Image Lab 6.0.1, Bio-Rad Laboratories, Inc., Hercules, CA, USA), and normalized to GAPDH [37,38,39,40]. 

### 2.6. Statistical Analysis

Data are expressed as the mean ± SEM. Statistical analysis was performed with two-tailed unpaired Student’s *t* test. A *p* value of less than 0.05 was considered statistically significant.

## 3. Results

### 3.1. HFHCD Induces NASH with Advanced Liver Fibrosis and Severe Atherosclerosis in Rabbits for 16 Weeks

We generated HFHCD containing palm oil 7.5%, cholesterol 0.5%, and ferrous citrate 0.5% and administered it to Japanese White rabbits for 16 weeks. Their livers were markedly enlarged and looked yellowish (Figure 1A). Histological findings of the liver revealed macrovesicular and microvesicular steatosis with massive pericellular fibrosis. Additionally, atherosclerotic plaques emerged in the aortic wall (Figure 1B). Serum AST, ALT, NEFA, TG, T-Chol and TBA levels were significantly increased in HFHCD-fed rabbits (Figure 1C). Hepatic contents of T-Chol and TBA were much higher in HFHCD rabbits compared to normal rabbits, and F-Chol also tended to increase; however, other lipid species such as NEFA, TG, and PL did not increase, presumably due to the progression of severe fibrosis in the liver. These results indicate that Japanese White rabbits fed HFHCD for 16 weeks exhibited pathologies resembling human NASH with advanced fibrosis and atherosclerosis.

### 3.2. Time–Course Study of HFHCD-fed Rabbits

We conducted the time–course study to elucidate the progression of pathology in HFHCD-fed rabbits. The liver-to-body weight ratio was gradually increased. The spleen-to-body weight also increased in HFHCD-fed rabbits, suggesting the advancement of liver fibrosis (Figure 2A). Hepatic histology revealed microvesicular steatosis around the central vein for a 4-week HFHCD treatment. In addition to microvesicular steatosis, hepatocyte ballooning and inflammatory cell infiltration were observed in the 8-week HFHCD group. Hepatocyte ballooning and infiltration of inflammatory cells exacerbated in the 14-week HFHCD group. HFHCD feeding caused mild perivenular fibrosis within 8 weeks, and extensive perivenular/pericellular fibrosis at the time of 14 weeks. Atherosclerotic plaques emerged in the aortic wall in the 8-week HFHCD group and extended to the entire circumference with the marked accumulation of foam cells in the 14-week HFHCD group (Figure 2B). The remarkable ballooned hepatocytes appeared in 14-week HFHCD rabbit livers, and the crystal structures were also observed in the hepatocytes (Figure 2C).

Serum AST and ALT levels in HFHCD group were 2–3-fold higher than in the control group at each time point. Serum lipid concentrations increased in the 4-week HFHCD group, especially T-Chol and TBA, in a time-dependent manner (Figure 2D). Hepatic lipid contents, including TBA, were markedly elevated in the 4-week HFHCD treatment. Although T-Chol, F-Chol, and TBA sustained higher levels at 14 weeks, NEFA, TG, and PL slightly decreased as time passed, resembling the pathological course of human NAFLD, called burned-out NASH (Figure 2E) [23,41]. These data indicate that HFHCD induced hepatic steatosis and hyperlipidemia within 4 weeks, emergence of hepatic fibrosis, and atherosclerotic plaque at 8 weeks, and the progression of advanced fibrosis and extensive atherosclerotic plaques at 14 weeks of HFHCD administration. 

### 3.3. HFHCD Alters Gene Expression Associated with Cholesterol Metabolism

A range of genes related to the lipid metabolism has changed in the liver of NAFLD/NASH patients [42,43]. To understand the mechanism of cholesterol accumulation in the liver during HFHCD feeding, we investigated the expression of several genes involved in the cholesterol metabolism. When excess cholesterol is present in the liver, sterol regulatory element-binding protein-2 (SREBP-2) is inactivated and its target gene-encoding HMG-CoA reductase (HMGCR), a rate-limiting enzyme of cholesterol synthesis, and LDLR are down-regulated. In the 4-week HFHCD treatment, *HMGCR* mRNA levels were decreased significantly, but returned to a similar level to the control group at 8 and 14 weeks. Other cholesterol synthesis-related gene encoding farnesyl-diphosphate farnesyltransferase 1 (*FDFT1*) was up-regulated only in the 4-week HFHCD group and opposed to squalene monooxygenase (*SQLE*). No significant changes were observed at the mRNA levels in 8 and 14 weeks of HFHCD group (Figure 3A). The gene expression associated with cholesterol uptake, the *LDLR* mRNA levels, were significantly decreased, whereas *CD36* and VLDL receptor (*VLDLR*) mRNA levels were increased by HFHCD feeding (Figure 3B). In genes related to hepatic cholesterol excretion, ATP-binding cassette subfamily A member 1 (*ABCA1*) mRNA levels were considerably increased by HFHCD in a time-dependent manner, while ATP-binding cassette subfamily G member 8 (*ABCG8*) mRNAs were increased slightly (Figure 3C). Other genes encoding apoB (APOB), microsomal triglyceride transfer protein (*MTTP*), and ATP-binding cassette subfamily G member 5 (*ABCG5*) were unchanged between HFHCD and control group. Although *LDLR* mRNA levels were decreased, a significant up-regulation of *CD36/VLDLR* mRNA expression is one of the causes of excessive cholesterol accumulation in the liver through the enhancement of the uptake of cholesterol from the circulation. 

### 3.4. HFHCD Disrupts BA Homeostasis in the Liver

A series of genes involved in BA synthesis and excretion also contributes to cholesterol metabolism. Cytochrome P450 7A1 (CYP7A1) is a rate-limiting enzyme of BA synthesis, while bile salt export pump (BSEP, encoded by *ABCB11*) is a major pump excreting BA from hepatocytes, and both genes are negatively and positively regulated by farnesoid X receptor (FXR). The gene expression of *CYP7A1* and *ABCB11* was not altered in the livers of HFHCD (Figure 3E). Similar to these data, the expression of other genes regulated by FXR, such as cytochrome P450 family 8 subfamily B member 1 (*CYP8B1*)*,* cytochrome P450 family 27 subfamily A member 1 (*CYP27A1),* and multidrug resistance-associated protein 2 (*MRP2*), was largely unchanged. The mRNA levels of solute carrier family 10 member 1 (*NTCP*) were down-regulated presumably due to enhanced hepatic inflammation and increased expression of tumor necrosis factor α (TNFα) and interleukin 1β (IL1-β) [44]. Since multiple drug resistance 1 (MDR1) acts as a cytoprotective function through the excretion of toxic compounds, the induction of MDR1 seemed to be a compensation to protect liver function. These data indicated that the regulation of FXR-based BA homeostasis was collapsed, and the accumulation of excess BA in the liver and the rise of serum BA levels occurred. 

Cholesterol ester is converted to F-Chol via neutral cholesterol ester hydrolase 1 (NCEH1), and the inverse reaction is performed by acetyl-CoA acetyltransferase 2 (ACAT2). Elevated levels of F-Chol in the HFHCD livers may involve increased *NCEH1* expression with unchanged *ACAT2* expression (Figure 3D). 

Overall, excessive accumulation of hepatic T-Chol and F-Chol is considered to be the result of an increased level of cholesterol uptake and dysregulation of BA homeostasis, the unsuitable expression of these cholesterol metabolic genes. 

### 3.5. HFHCD Does Not Affect the Expression of Genes Related to Fatty Acid Metabolism

Acetyl-CoA carboxylase encoded by *ACACA* gene is a rate-limiting enzyme of long-chain fatty acid synthesis. The mRNA levels of *ACACA* were upregulated in the livers of the HFHCD group. In contrast, the expression of diacylglycerol acyltransferase 2 (*DGAT2*) involved in the final stage of TG synthesis was decreased (Figure 4A). Concerning fatty acid uptake, fatty acid-binding protein 1 (*FABP1*) mRNA tended to decrease, while hepatic triglyceride lipase (*HTGL*) increased in HFHCD group. Long-chain fatty acid CoA ligase 1 (*ASCL1*) mRNA was elevated only in the 4-week HFHCD group (Figure 4B). No substantial changes in the expression of genes related to fatty acid oxidation were observed in the HFHCD livers (Figure 4C). 

### 3.6. HFHCD Up-Regulates Gene Expression Related to Inflammation and Fibrosis

Since HFHCD caused NASH with fibrosis, we investigated the expression of the genes associated with inflammation and ensuing fibrogenesis. The mRNA levels of macrophage marker *CD68* and the inflammatory cytokines, such as *TNFα*, C-C motif chemokine ligand 2 (*CCL2*), *IL1-β*, and interleukin 18 (*IL18*), were significantly increased in the HFHCD livers (Figure 5A). In addition to transforming growth factor β1 (*TGFB1),* an inducer of fibrosis, the gene expression of the fibrogenic factors, such as actin alpha 2 (*ACTA2*), collagen type I alpha 1 chain (*COL1A1*), and collagen type III alpha 1 chain (*COL3A1*), was elevated in the HFHCD livers (Figure 5D). The mRNA levels of toll-like receptors (TLRs) and their adaptor (*TLR2*, *TLR4, and CD14*) and inflammasome components, such as NLR family pyrin domain containing 3 (*NLRP3*) and caspase 1 (*CASP1*), were also induced (Figure 5B,C). 

F-Chol causes ER stress and the resulting oxidative stress, stimulating TLRs and inducing inflammation. Cholesterol crystals activate the NLRP3 inflammasome and contribute to the inflammatory signaling. ER stress-related genes, such as PRKR-like endoplasmic reticulum kinase (*PERK*) and *DDIT3* and ER stress-induced lipoapoptosis, such as Bcl-2-like protein 11 (*BIM*) and p53 upregulated modulator of apoptosis (*PUMA*), were increased (Figure 6A). Furthermore, oxidative stress-producing genes (cytochrome b-245 beta chain, *CYBB*; and neutrophil cytosolic factor 1, *NCF1*) were increased in the HFHCD group (Figure 6B), but oxidative stress-eliminating genes, NAD(P)H quinone dehydrogenase 1 (*NQO1*), heme oxygenase 1 (*HMOX1*), and superoxide dismutase 1 (*SOD1*) were up-regulated as well, and the amount of the 4-hydroxy-nonenal (4-HNE), a marker of the lipid peroxide, was not increased (Figure 6C). These data demonstrate that cholesterol accumulation in the liver and ensuing ER stress, inflammasome activation and enhanced TLR signaling triggers hepatic inflammation and fibrosis progression in this rabbit NASH model. 

## 4. Discussion

We developed the original HFHCD containing palm oil 7.5%, cholesterol 0.5%, and ferrous citrate 0.5% and administered it to Japanese White rabbits, which have close similarities to lipoprotein and BA metabolism in humans. We found the progression of NASH with advanced fibrosis and atherosclerosis at 14-weeks of HFHCD treatment. Investigation of hepatic lipid profiles and gene expression revealed the significance of hepatic cholesterol/BA accumulation and ER stress to the pathogenesis of NASH in this model. Our model may be useful to evaluate the pathogenesis of NAFLD/atherosclerosis and seek concurrent treatment for both diseases.

Several animal models have been used to evaluate the pathogenesis of NAFLD/NASH. Mice and rats are the most common due to their easy handling and genetic manipulation, and short life cycles. However, there are some great differences between these animals and humans, and differences in lipid metabolism seem to be one of the most crucial factors as regards considering translatability of subclinical studies to clinical trials. Indeed, mice and rats have high circulating HDL and low LDL, which is opposite to humans. Rodents have both apoB-48 and -100 in VLDL particles, and humans only have apoB-100. VLDL particles including apoB-48 have a more rapid fractional catabolic rate than VLDL particles that only have apoB-100. Additionally, BA composition, a key determinant for cholesterol homeostasis, is quite different between rodents and humans as well. Cholic acid (CA) and chenodeoxycholic acid (CDCA) are synthesized as primary BA in humans, while mice and rats have CYP2C70, which converts CDCA to muricholic acid (MCA); therefore, approximately half of the BA pool of mice is MCAs. Although CDCA has the highest affinity for FXR, MCA is antagonistic to FXR. Therefore, the BA profile may affect the high synthetic rate of BA in mice and the resulting rapid cholesterol metabolism. The RCT system is also different because rodents have no cholesterol ester transfer protein (CETP) activity in plasma, which contributes to the high HDL and low LDL levels. It is known that statin’s effect is not shown on mice, possibly due to the high synthetic capacity of cholesterol [28]. According to the abovementioned reasons, choosing animals whose lipid metabolism resembles that of humans as closely as possible is desired. Rabbits have more similarities in lipoprotein metabolism with that of humans compared with rodents. Plasma LDL levels are high and HDL levels are low due to CETP activity [28], and the BA composition is predominantly hydrophobic and FXR agonistic [30].

Our HFHCD-fed rabbits represented typical pathological features of NASH, such as hepatic steatosis, pericellular fibrosis, and hepatocyte ballooning, and clinical features, including hypertransaminasemia, cholesterol-dominant hyperlipidemia, and accompanying atherosclerosis. It is reported that macrophages in atherogenic lesions in humans and rabbits have VLDLR but do not have it in mice [45,46]. Although our model did not show obesity and hyperglycemia, key contributors to NAFLD/NASH, it was documented that non-obese NAFLD men were prone to coronary artery calcification [47]. Therefore, this rabbit model seems to be a suitable animal to evaluate human NAFLD/NASH, especially with hypercholesterolemia and atherosclerosis, and assess the efficacy of therapeutic interventions on NASH/atherosclerosis. 

Cholesterol, especially F-Chol, plays a critical role in the development of NAFLD and progression to NASH [48,49,50]. The imbalance of F-Chol content in the ER membrane causes ER stress, and chronic ER stress results in inflammation and cell death. When an excessive amount of F-Chol accumulates in the cells, a cholesterol crystal is formed. Crystalized cholesterol is detected in the hepatocytes of NASH patients. Crystalized cholesterol activates NLRP3 inflammasome, inducing hepatocyte death [49,51]. Subsequently, Kupffer cells are activated and release the proinflammatory cytokines, and hepatic stellate cells are activated as well, leading to hepatic inflammation and fibrosis. 

In this HFHCD-fed rabbit model, the hepatic expression levels of some genes involved in cholesterol metabolism were markedly changed. The mRNA levels of CD36 and VLDLR were increased via HFHCD. These genes are normally expressing at very low levels in the hepatocytes, but CD36 expression increases in NASH patients and VLDLR increases in some pathological conditions, such as obesity, diabetes, and NAFLD [52]. Changes in the expression of these genes could contribute to the elevated absorption of cholesterol into the liver. 

Greater accumulation of F-Chol in the liver prompted us to speculate about the alteration of the cholesterol-esterizing pathway in this HFHCD-fed rabbit model. F-Chol is generated from cholesterol ester via hydrolysis with NCEH1, while ACAT2 plays the reverse function of esterifying F-Chol. The balance of intracellular free and esterified cholesterol is preserved by both genes. In HFHCD livers, *NCEH1* mRNA significantly increased, but *ACAT2* was unchanged. Since hepatic NCEH1 is reportedly up-regulated in NAFLD patients [53], this model is considered to be more appropriate for investigating the contribution of F-Chol to the pathogenesis of NASH. 

Excess amount of F-Chol is excreted in the bile by ABCG5 and ABCG8; however, in the HFHCD-fed livers, the gene expression of both genes was not significantly changed. ABCA1 is involved in the production of HDL and thus contributes to the extracellular excretion of cholesterol. ABCA1 is one of the target genes of the liver X receptor (LXR) the ligand of which is oxysterol, oxidized derivatives from cholesterol. Excessive levels of F-Chol are present in the liver of HFHCD-fed rabbits, and significant amounts of oxysterol are expected. Activated LXR induces ABCA1 gene expression and promotes cholesterol efflux from the hepatocyte. This effect is regarded as the protective effect from the F-Chol toxicity.

The strong points of this model are similarities of disease phenotypes of dyslipidemic NAFLD/NASH patients with accompanying atherosclerosis, and relatively speedy progression of liver fibrosis and atherogenic plaques through singular HFHCD for 8–14 weeks. Kim et al. documented that the treatment of New Zealand white rabbits with a high-cholesterol diet (1%) for 3 months led to the rabbits exhibiting mild fatty changes in the liver with sporadic fibrosis [54]. Such a difference might stem from differences in diet composition and/or the genetic background of rabbits. Considering the speedy progression of NASH fibrosis/atherosclerosis and similarities in systemic cholesterol/lipoprotein metabolism and VLDLR expression in macrophages of atherogenic plaques to humans, utilizing this rabbit model would save time to assess NASH fibrosis/atherosclerosis-retarding effects of candidate agents, such as cholesterol-lowering statins and ezetimibe. 

Recently, the association between hepatic zonation and NAFLD/NASH development has been gathering attention [55]. Differences in hepatic zone-dependent enzyme expression and nutrient metabolism between humans and rodents might be associated with the fact that mouse models of NAFLD/NASH are unable to replicate the human disease [24,56]. To date, rabbit liver zonation has not been fully investigated. If hepatic zonation is more similar to humans compared with that of rodents, this finding would corroborate the relevance of using rabbit models in NAFLD/NASH research. 

## 5. Conclusions

We demonstrated a new rabbit NASH model using Japanese White rabbits and a newly developed HFHCD. Although further studies are needed to validate the usefulness of this model, it is expected that this rabbit NASH model may contribute to the development of favorable treatments for NASH, liver fibrosis, and atherosclerosis. 

## Figures and Tables

**Figure 1 biomedicines-11-00384-f001:**
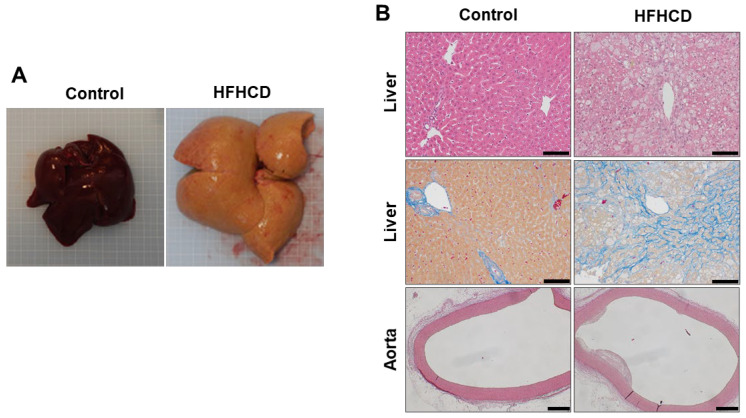
HFHCD induced advanced liver fibrosis and hyperlipidemia in rabbits. (**A**) Liver tissue of the control and 16-week HFHCD groups. (**B**) Histochemical staining of the liver and aorta of the control and 16-week HFHCD group; HE staining of the liver (upper); scale bar = 100 µm, Azan–Mallory staining (middle); scale bar = 100 µm, HE staining of the aorta (bottom); scale bar = 400 µm. (**C**) Serum biochemical profiles of the control and HFHCD-fed rabbits. (**D**) Hepatic lipid and bile acid profiles of the control and 16-week HFHCD groups. Data are shown as the mean ± SEM. * *p* < 0.05, ** *p* < 0.01, and *** *p* < 0.001 vs. control group. Control group (n = 3), HFHCD group (n = 3).

**Figure 2 biomedicines-11-00384-f002:**
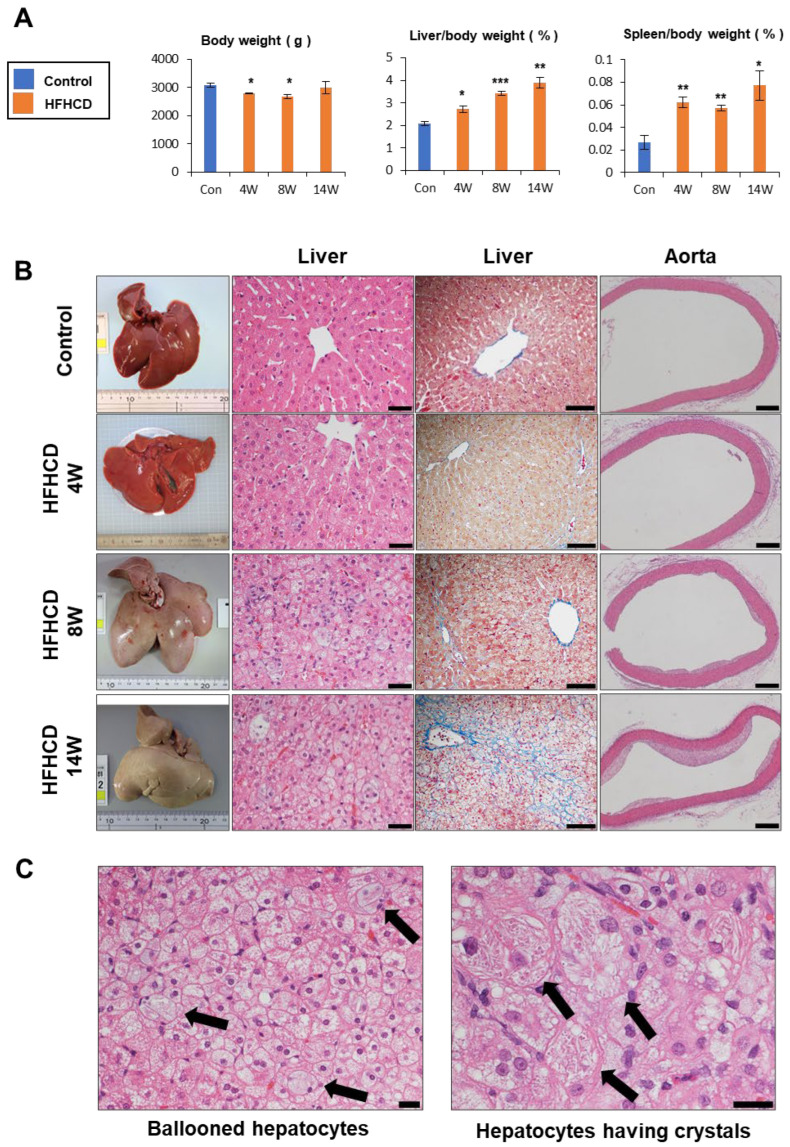
Time–course study of HFHCD-fed rabbits. (**A**) Body weight and liver-to-body weight, spleen-to-body weight ratio. (**B**) Liver tissue and histochemical staining of the liver and aorta of the control and HFHCD-fed rabbits; liver HE staining (**left**); scale bar = 40 µm, liver Azan–Mallory staining (**middle**); scale bar = 100 µm, aorta HE staining (**right**); scale bar = 400 µm. (**C**) Ballooned hepatocyte and crystal structures in 14-week HFHCD-fed rabbit livers. Scale bar = 20 µm. (**D**) Serum biochemical profiles. (**E**) Hepatic lipid and bile acid profiles. Data are shown as the mean ± SEM. * *p* < 0.05, ** *p* < 0.01, and *** *p* < 0.001 vs. control group. Control group (12 weeks, n = 3), HFHCD group; 4 weeks (n = 3), 8 weeks (n = 4), 14 weeks (n = 4).

**Figure 3 biomedicines-11-00384-f003:**
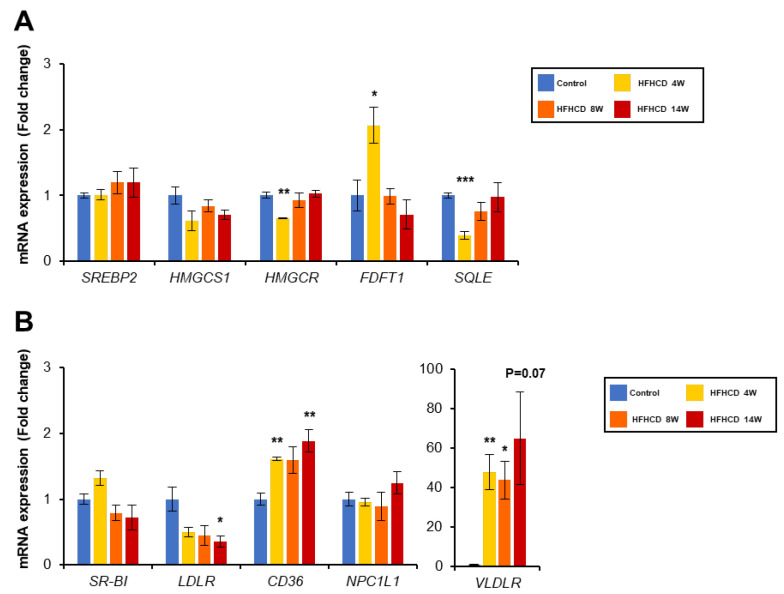
HFHCD altered the gene expression of cholesterol metabolism. Hepatic mRNA levels of genes related to cholesterol synthesis (**A**), cholesterol uptake to the liver (**B**), cholesterol excretion (**C**), cholesterol esterification and hydroxylation (**D**), and bile acid metabolism (**E**) were quantified using real-time PCR and normalized to GAPDH. Data are shown as the mean ± SEM. * *p* < 0.05, ** *p* < 0.01, and *** *p* < 0.001 vs. control group. Control group (12 weeks, n = 3), HFHCD group; 4 weeks (n = 3), 8 weeks (n = 4), 14 weeks (n = 4).

**Figure 4 biomedicines-11-00384-f004:**
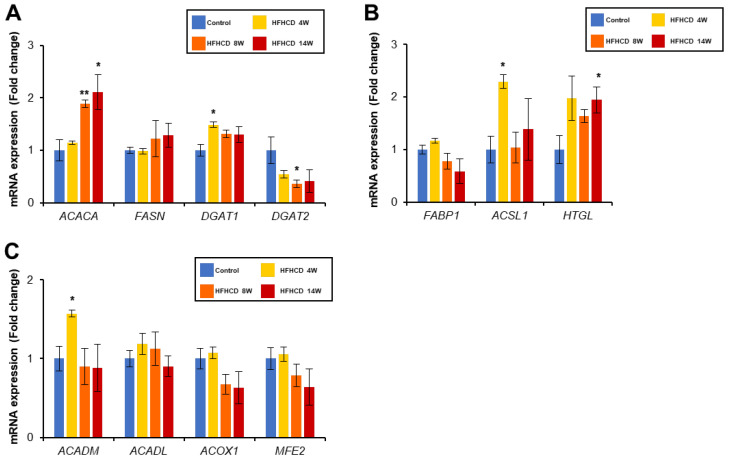
The changes in the gene expression of TG metabolism in HFHCD-fed rabbit livers. Hepatic mRNA levels of genes related to fatty acid/TG synthesis (**A**), TG transport (**B**), and β-oxidation (**C**) were quantified using real-time PCR and normalized to GAPDH. Data are shown as the mean ± SEM. * *p* < 0.05, ** *p* < 0.01 vs. control group. Control group (12 weeks, n = 3), HFHCD group; 4 weeks (n = 3), 8 weeks (n = 4), 14 weeks (n = 4).

**Figure 5 biomedicines-11-00384-f005:**
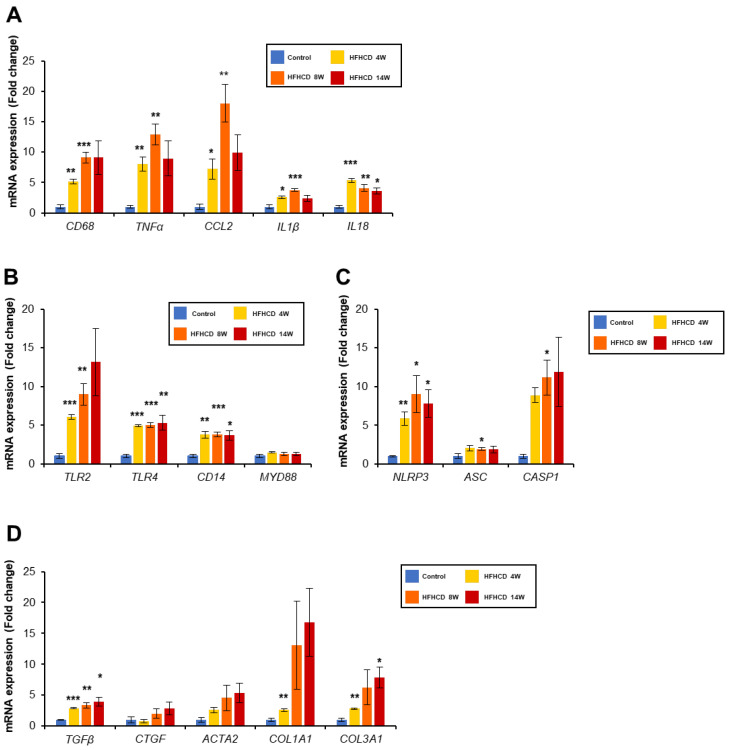
HFHCD increased the inflammatory and fibrogenic genes. Hepatic mRNA levels of genes related to inflammation (**A**), TLRs (**B**), NLRP3 inflammasome (**C**), and fibrosis (D) were quantified via real-time PCR and normalized to GAPDH. Data are shown as the mean ± SEM. * *p* < 0.05, ** *p* < 0.01, and *** *p* < 0.001 vs. control group. Control group (12 weeks, n = 3), HFHCD group; 4 weeks (n = 3), 8 weeks (n = 4), 14 weeks (n = 4).

**Figure 6 biomedicines-11-00384-f006:**
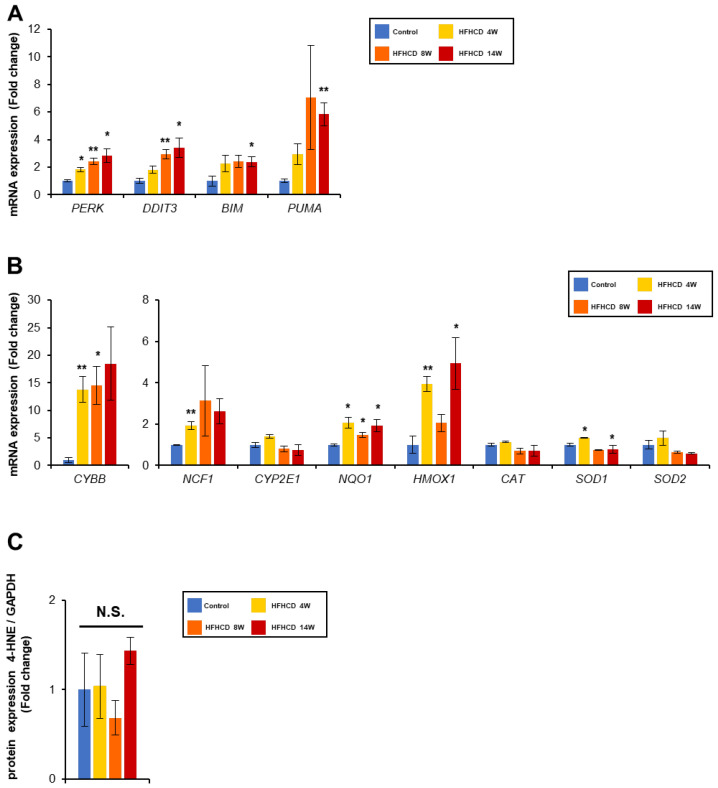
HFHCD induced ER stress and related apoptosis genes, but did not affect oxidative stress. (**A**,**B**) Hepatic mRNA levels of genes related to ER stress (**A**), oxidative stress production and elimination (**B**) were quantified via real-time PCR and normalized to GAPDH. (**C**) Hepatic protein levels of 4-HNE were determined using immunoblot analysis. The band intensity was normalized to that of GAPDH and expressed as a relative value compared with control rabbits. The full-length blots are shown in Appendix A. Data are shown as the mean ± SEM. * *p* < 0.05, ** *p* < 0.01 vs. control group. N.S., not significant. Control group (12 weeks, n = 3), HFHCD group; 4 weeks (n = 3), 8 weeks (n = 4), 14 weeks (n = 4).

## Data Availability

Not applicable.

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
