# Peer review of "Development of the Rabbit NASH Model Resembling Human NASH and Atherosclerosis"

_biomedicines, 2023, doi:10.3390/biomedicines11020384_

Round 1

Reviewer 1 Report

Obesity is a risk factor for complications of atherosclerotic vascular disease such as myocardial infarction and stroke. Recent studies have demonstrated that the vascular risk associated with obesity is correlated particularly with visceral adiposity.Visceral fat has been demonstrated to express more inflammatory cytokines than subcutaneous fat in obese states….as evident in….Visceral adipose tissue and atherosclerosis. Curr Vasc Pharmacol. 2009 Apr;7(2):169-79. doi: 10.2174/157016109787455680. PMID: 19356000.

Anatomically adjacent to the liver, visceral adipose tissue  plays an important role in NAFLD pathogenesis via its diabetogenic, atherogenic, and pro-inflammatory functions. Many anthropometric metrics and insulin-resistance markers are being studied to gauge visceral adipose tissue  function, with the goal of predicting metabolic risk and NAFLD surveillance…as evident in….Nonalcoholic Fatty Liver Disease: The Role of Visceral Adipose Tissue. Clin Liver Dis (Hoboken). 2022 Jan 28;19(3):106-110. doi: 10.1002/cld.1183. PMID: 35355841; PMCID: PMC8958250….and ….Is visceral fat reduction necessary to favour metabolic changes in the liver? J Gastrointestin Liver Dis. 2012 Jun;21(2):205-8. PMID: 22720311.

As visceral adipose tissue plays an unmistakable role in NASH and other obesity-related conditions, including cardiovascular disease and DM2, the lack of evaluating this aspect should certainly be considered a shortcoming…how this new proposed rabbit model overcome it?  

This, to assure a sure reproducibility, confronting the author’s model with another one, i.e., transgenic mice overexpressing 11beta HSD-1 selectively in adipose tissue to an extent similar to that found in adipose tissue from obese humans…as evident in…A transgenic model of visceral obesity and the metabolic syndrome. Science. 2001 Dec 7;294(5549):2166-70. doi: 10.1126/science.1066285. PMID: 11739957.

What is the main question addressed by the research?

Answer:This rabbit NASH model might contribute to exploring the concurrent treatment options for human NASH and atherosclerosis.

2. Do you consider the topic original or relevant in the field? Does it 

address a specific gap in the field?

Answer: The topic is extremely relevant in the field of NAFLD/NASH because opens the door to new therapeutical strategies

3. What does it add to the subject area compared with other published material?

Answer: The rabbit model  will clarify inner mechanisms underlying liver fibrosis  

4. What specific improvements should the authors consider regarding the 

methodology? What further controls should be considered?

Answer: Being a pilot study in which this model is proposed, it is necessary to look at the future application before adjusting the model, in case it does not fit the  aim.

5. Are the conclusions consistent with the evidence and arguments presented 

and do they address the main question posed?

Answer: The conclusions are consistent with the findings of the study 

6. Are the references appropriate?

Answer: The references are for the major part appropriate and up-to-date, but not complete.

7. Please include any additional comments on the tables and figures.

Answer: Both Tables and Figures are clear and very useful.

Author Response

Reviewer 1

Obesity is a risk factor for complications of atherosclerotic vascular disease such as myocardial infarction and stroke. Recent studies have demonstrated that the vascular risk associated with obesity is correlated particularly with visceral adiposity. Visceral fat has been demonstrated to express more inflammatory cytokines than subcutaneous fat in obese states….as evident in….Visceral adipose tissue and atherosclerosis. Curr Vasc Pharmacol. 2009 Apr;7(2):169-79. doi: 10.2174/157016109787455680. PMID: 19356000.

Anatomically adjacent to the liver, visceral adipose tissue plays an important role in NAFLD pathogenesis via its diabetogenic, atherogenic, and pro-inflammatory functions. Many anthropometric metrics and insulin-resistance markers are being studied to gauge visceral adipose tissue function, with the goal of predicting metabolic risk and NAFLD surveillance…as evident in….Nonalcoholic Fatty Liver Disease: The Role of Visceral Adipose Tissue. Clin Liver Dis (Hoboken). 2022 Jan 28;19(3):106-110. doi: 10.1002/cld.1183. PMID: 35355841; PMCID: PMC8958250….and ….Is visceral fat reduction necessary to favour metabolic changes in the liver? J Gastrointestin Liver Dis. 2012 Jun;21(2):205-8. PMID: 22720311.

As visceral adipose tissue plays an unmistakable role in NASH and other obesity-related conditions, including cardiovascular disease and DM2, the lack of evaluating this aspect should certainly be considered a shortcoming…how this new proposed rabbit model overcome it?  

This, to assure a sure reproducibility, confronting the author’s model with another one, i.e., transgenic mice overexpressing 11beta HSD-1 selectively in adipose tissue to an extent similar to that found in adipose tissue from obese humans…as evident in…A transgenic model of visceral obesity and the metabolic syndrome. Science. 2001 Dec 7;294(5549):2166-70. doi: 10.1126/science.1066285. PMID: 11739957.

Response

Thank you for your critical comments. We entirely agree with you regarding the importance of adipocyte hypertrophy/dysfunction in the pathogenesis of NAFLD/NASH. Recently, the heterogeneity of clinical features of NAFLD/NASH patients has been gathering attention. It was documented that non-obese NAFLD/NASH men were prone to coronary artery calcification (Sci Rep 2020,  #47), indicating the close association between non-obesity, NAFLD/NASH, and atherosclerosis. We are considering that our rabbit model may be useful for assessing the efficacy of therapeutic interventions on NASH/atherosclerosis, as sometimes observed in non-obese individuals. We added these descriptions (line 640-645) and the related references (#47) to the Discussion section.

Additional comments to Reviewer 1

We added two authors, because they also conducted animal treatment and experiments.

We added a full blot of 4-HNE-related proteins to Supplementary information, in order to guarantee the data transparency and certainty.

Thank you for reviewing our manuscript.

Reviewer 2 Report

The Authors deeply analysed the onset and progression of NAFLD and NASH in the liver and atherosclerosis histological changes in the aorta of Japanese rabbit placed on a High Fat-High cholesterol diet from 4 to 14 weeks.

This study is technically well conducted and interesting, even if, in my opinion, some minor changes are required to ameliorate the text as follows:

Introduction-The Authors focused on the importance of rabbit versus rodent models of NAFLD/NASH dietary induced and cholesterol transport. However, they failed to introduce crucial mechanisms involved in the pathogenesis of this disease like inflammation and ER stress-apoptosis reaction even then they analysed some mRNA markers in Results. Moreover, details of atherosclerotic plaque composition and foam cells must be indicated in brief sentences here. Please insert also more References on the above indications.

Results-The extent of steatosis, histologically detected at 4 weeks of dietary treatment, might be related to liver zonation. The indication of different zones in the liver is crucial to influence different metabolic activity. Please discuss zonation and eventually insert relative References (zone 1 Periportal versus zone 3 Pericentral area). See Cunningham and Porat-Shliom Front Physiol 12, 732929, 2021.

Author Response

Reviewer 2

The Authors deeply analysed the onset and progression of NAFLD and NASH in the liver and atherosclerosis histological changes in the aorta of Japanese rabbit placed on a High Fat-High cholesterol diet from 4 to 14 weeks. This study is technically well conducted and interesting, even if, in my opinion, some minor changes are required to ameliorate the text as follows:

  1. Introduction-The Authors focused on the importance of rabbit versus rodent models of NAFLD/NASH dietary induced and cholesterol transport. However, they failed to introduce crucial mechanisms involved in the pathogenesis of this disease like inflammation and ER stress-apoptosis reaction even then they analysed some mRNA markers in Results. Moreover, details of atherosclerotic plaque composition and foam cells must be indicated in brief sentences here. Please insert also more References on the above indications.

Response

We deeply appreciate your critical comments. According to your instructions, we added the descriptions regarding the contribution of ER stress, inflammasome, lipoapoptosis and pyroptosis to NASH development and the related references (#7-10) to the Introduction section. Additionally, we mentioned the detailed composition of atherogenic plaques and differences in plaque pathology between rabbits and mice in the Introduction section and added the related references (#31-35).

  1. Results-The extent of steatosis, histologically detected at 4 weeks of dietary treatment, might be related to liver zonation. The indication of different zones in the liver is crucial to influence different metabolic activity. Please discuss zonation and eventually insert relative References (zone 1 Periportal versus zone 3 Pericentral area). See Cunningham and Porat-Shliom Front Physiol 12, 732929, 2021.

Response

Thank you for your important suggestion. Indeed, the fact that mouse NAFLD/NASH models cannot completely reproduce human pathology might stem from difference in metabolism associated with liver zonation, such as location of lipogenesis and detoxification. Metabolic zonation in rabbit livers has not been investigated yet, but if it would be more similar to humans compared with rodents, this finding would corroborate the relevance to use rabbit models for evaluating human NAFLD/NASH pathogenesis. We added these descriptions and the related references (#55-57, including Cunningham's manuscript) to the Discussion section.

Additional comments to Reviewer 2

We added two authors, because they also conducted animal treatment and experiments.

We added a full blot of 4-HNE-related proteins to Supplementary information, in order to guarantee the data transparency and certainty.

Thank you for reviewing our manuscript.